# Simple Method for the Determination of THC and THC-COOH in Human Postmortem Blood Samples by Gas Chromatography—Mass Spectrometry

**DOI:** 10.3390/molecules28083586

**Published:** 2023-04-20

**Authors:** Ivan Álvarez-Freire, Anxa Valeiras-Fernández, Pamela Cabarcos-Fernández, Ana María Bermejo-Barrera, María Jesús Tabernero-Duque

**Affiliations:** Forensic Toxicology Service, Institute of Forensic Sciences, Faculty of Medicine, University of Santiago de Compostela, 15782 Santiago de Compostela, Spain

**Keywords:** THC/THC-COOH, GC-MS quantitation, postmortem blood, liquid-liquid extraction

## Abstract

A simple and sensitive analytical method was developed for qualitative and quantitative analysis of Δ9-tetrahydrocannabinol (Δ9-THC) and its metabolite 11-nor-Δ9-tetrahydrocannabinol-carboxylic acid (Δ9-THC-COOH) in human postmortem blood using gas chromatography/mass spectrometry (GC-MS) in selected ion monitoring (SIM) mode. The method involved a liquid-liquid extraction in two steps, one for Δ9-THC and a second one for Δ9-THC-COOH. The first extract was analyzed using Δ9-THC-D3 as internal standard. The second extract was derivatized and analyzed using Δ9-THC-COOH-D3 as internal standard. The method was shown to be very simple, rapid, and sensitive. The method was validated for the two compounds, including linearity (range 0.05–1.5 µg/mL for Δ9-THC and 0.08–1.5 µg/mL for Δ9-THC-COOH), and the main precision parameters. It was linear for both analytes, with quadratic regression of calibration curves always higher than 0.99. The coefficients of variation were less than 15%. Extraction recoveries were superior to 80% for both compounds. The developed method was used to analyze 41 real plasma samples obtained from the Forensic Toxicology Service of the Institute of Forensic Sciences of Santiago de Compostela (Spain) from cases in which the use of cannabis was involved, demonstrating the usefulness of the proposed method.

## 1. Introduction

Cannabis is one of the longest-established drugs in Europe. Internationally and within Europe, cannabis use continues to be a topic that is generating significant policy and public interest, as new developments are triggering debate on how society should respond to this substance. New forms of cannabis have been developed in recent years as a result of advances in cultivation, extraction and production techniques. The creation of legal recreational cannabis markets where the drug has been legalized is also driving innovation, with the development of new cannabis products such as edibles, e-liquids, and concentrates. Some of these are now appearing on the European market, where they represent a new challenge for detection and drug control [1].

The European Drug Report 2022: Trends and Developments from European Monitoring Centre for Drugs and Drug Addiction (EMCDDA) says that around 83.4 million or 29% of adults (aged 15–64) in the European Union are estimated to have tried illicit drugs during their lives. The most consumed drug is cannabis (with over 22 million European adults reporting its use in the last year). Levels of lifetime use of cannabis differ considerably between countries, ranging from around 4.3% of adults in Malta to 44.8% in France. Across all age groups, cannabis is the most used illicit drug. The drug is generally smoked, and, in Europe, it is commonly mixed with tobacco. Currently, the average Δ9-THC content of resin (21%) is almost twice that of herbal cannabis (around 11%). This percentage indicates a variation, since until a few years ago, the Δ9-THC content of cannabis herb was higher than that of the resin. Additionally, patterns of cannabis use can range from occasional use to the regular and dependent use [2].

Until a few years ago, the traffic in both cannabis herb and resin came from the Western Balkans and Morocco, respectively. This trend underwent a major change due to the mobility restrictions caused by COVID-19. It currently appears that domestically produced cannabis has become the most important source for the European market (2020 likedata). Reports on the increase in seizures of cannabis herb in Spain suggest that our country may become the main supplier of the EU market [2].

In recent years, the consumption of this substance has been associated with the presence of multiple psychosocial problems in young people. Several studies have associated cannabis use with problems such as dropping out of school and declining academic performance [3], risky sexual practices [4], or excessive alcohol consumption, among other things [5]; likewise, it is also frequently detected in the body of drivers arrested for erratic driving or involved in traffic accidents.

The main psychoactive constituent of cannabis is tetrahydrocannabinol (Δ9-THC), also known by its international common name as dronabinol. This substance comes from a plant known as *Cannabis sativa* L., which is widely distributed, growing preferentially in temperate and tropical zones. The psychoactive and medicinal chemical compounds found in the resin of this plant are known as cannabinoids. The cannabis plant contains more than 460 known compounds [6]; over 70 of these have a cannabinoid structure, including carboxylic acids, analogs, and degradation products. Depending on their chemical structure cannabinoids are divided into several sub-categories. Δ9-tetrahydrocannabinol is primarily responsible for the psychoactive effects of cannabis but is present in the cannabis plant to a major extent as a mixture of its non-psychoactive precursors Δ9-tetrahydrocannabinolic acids A and B (Δ9-THCA and Δ9-THCB). In blood and other body compartments, the metabolites 11-hydroxy-THC (THC-OH), 11-nor-9-carboxy-THC (THC-COOH), and the acyl glucuronide of Δ9-THC-COOH are also detected in greater abundance [7]. Δ9-tetrahydrocannabinol is rapidly assimilated after exposure and, because of its lipophilic nature, is distributed to adipose tissue, liver, lung, and spleen. It is then slowly released back into the blood and metabolized, causing a relatively long terminal half-life. Hepatic enzymatic biotransformation yields multiple metabolites with C-11 being the major modification site. Hydroxylation results in the psychoactive compound 11-hydroxy-Δ9-tetrahydrocannabinol (11-OH-THC) and further oxidation in the inactive 11-nor-Δ9-tetrahydrocannabinol-9-carboxylic acid (Δ9-THC-COOH), the most important compound for testing purposes, which is excreted into the urine mainly as a glucuronic acid conjugate [8].

Pharmacokinetic data of cannabinoids were extensively reviewed. Peak concentrations for Δ9-THC were observed approximately 8 (range, 6–10) minutes after onset of smoking, whereas that of Δ9-THC-COOH was at 81 (range, 32–133) minutes. The mean window of detection for Δ9-THC-COOH in plasma samples was 3.5 (range, 2–7) days after smoking a cigarette containing 16 mg of Δ9-THC and 6.3 (range, 2–7) days after a 34 mg dose. In 52 volunteers admitted to a detoxification, Δ9-THC-COOH could be detected in serum from 3.5 to 74.3 h [8].

Due to the frequency of cannabis abuse and the increase in the consumption of other cannabinoids as therapeutic agents (for cancer and pain treatment), or in cosmetic products and foods, it is necessary to find the most suitable detection methods for the determination of Δ9-THC and its metabolites in different biological matrices [9].

Therefore, in the toxicological determination of cannabis, it is important to consider not only the use of cannabis as a drug of abuse, but also its legal use. While the former is high in Δ9-THC, legal cannabis contains mostly CBD (and less than 1% Δ9-THC); however, it can reach detectable concentrations in blood. For this reason, the determination of 11-nor-9-carboxy-THC (Δ9-THC-COOH) in forensic toxicology laboratories is of special interest, to distinguish between the use of legal cannabis (mainly with CBD) and that of cannabis as a drug of abuse (mainly Δ9-THC, from which the inactive metabolite Δ9-THC-COOH is derived). This is especially important in countries where the use of high-CBD, low-THC cannabis products is legal to verify drug withdrawal [10].

In response to the growing demand for reliable evidence of cannabis use, several procedures have been developed to determine the presence of Δ9-THC and/or its metabolites in body fluids, such as oral fluid [11,12], urine [11], plasma [13,14,15], whole blood [11,16,17,18,19,20,21,22,23], umbilical cord [24], hair [25,26], breath [21], and meconium [27,28] involving different analytical techniques such as GC-MS [13,18,20,28], GC-MS/MS [11,15,26,27,29], LC-MS/MS [16,17,18,19]. Many of these methods use SPE as extraction procedure; however, in a forensic toxicology laboratory, LLE is sometimes a more appropriate method because it allows working with postmortem blood directly. Sometimes the blood degrades and introducing it into the SPE cartridges causes them to get stuck, making extraction difficult or impossible.

The present work proposes a simple and fast method for the determination of Δ9-THC and Δ9-THC-COOH in postmortem blood and its usefulness has been demonstrated by applying it to 41 real postmortem samples received in our Forensic Toxicology Service.

## 2. Results and Discussion

The analysis of cannabinoids becomes important if the presence of cannabis is suspected to have played a role in the cause of death (for example, in workplace accidents or fatal traffic accidents). Determining whether a person was affected by cannabis use is not straightforward due to the difficulty of establishing whether postmortem blood concentrations can be correlated with pre-death status. Drug detection time (the time after drug administration when it is still detectable) is an important factor that must be considered in the analysis of drug content in biological fluids. Δ9-THC concentrations in plasma or other biological fluids will depend on pharmacological factors (e.g., drug dose, origin of the plant, way the drug is prepared, route of administration, and rates of metabolism and excretion) and analytical factors (e.g., sensitivity, specificity, and accuracy of the analytical method) [30].

In chronic cannabis smokers, the finding of Δ9-THC at low blood concentrations has been described up to 30 days after consumption [31,32]. The most frequent way of consuming cannabis in our environment is smoking cigarettes or pipes, mixed with tobacco, although in recent years other ways of use have proliferated, such as vaporizers, oils or oral intake. These new preparations and forms of consumption also poses new challenges to identifying cannabis-derived compounds in biological fluids.

Traditionally, Δ9-THC and its non-psychoactive metabolite Δ9-THC-COOH were the primary analytes monitored for cannabis use laboratories in suspected driving under drug influence cases, drugs in the workplace, criminal justice, and substance use controls in the field of pain management because smoking was the primary route of administration.

Δ9-THC blood concentrations decline rapidly after inhalation: approximately 74% at 30 min and 90% at 1.4 h. Δ9-THC-COOH and its phase II glucuronidation metabolites (the primary non-psychoactive metabolites of Δ9-THC) report a previous use of cannabis and allow for an increase in the detection window since their elimination half-lives are much longer compared to Δ9-THC and 11-OH-THC [33].

Liquid-liquid extraction is quick, efficient, and often more favourable for postmortem blood due to the nature of the sample matrix [20,34]. Blood plasma is generally considered the most useful sample for drug identification in quantitative analyses; likewise, given that psychoactive substances that are tested often leave the blood rapidly, this sample is most useful for the purpose of identifying the recent use of drugs.

The selection of internal standards is an important factor in the development of quantitative assays involving MS. Due to the demand for effective internal standards for MS analysis of Δ9-THC and its major metabolites, a variety of deuterium-labelled analogues have been commercialized [35]. In our case we have opted for the ones with three deuterium atoms.

The method developed allowed the determination of the analytes with good sensitivity. The derivatization step was necessary in order to improve the sensitivity for Δ9-THC-COOH, which showed poor chromatographic behavior and a lack of sensitivity if injected underivatized.

LOD and LLOQ are shown in Table 1. Our values are better than those obtained by other authors [19,29]. The method was linear in the range with quadratic regression coefficients of 0.992 (Δ9-THC) and 0.991 (Δ9-THC-COOH) (Table 1).

The repeatability of concentrations and accuracy were acceptable for all the substances (coefficients of variation (CV) of concentration values and mean analytical error were lower than 15% for all the compounds studied, both for intraday and interday experiments). Results from the validation study are summarized in Table 2.

The analyses performed on 10 negative samples did not show significant interferences at the retention times of the analytes. This confirms that the method possesses adequate selectivity. The recovery data obtained demonstrate that the extraction procedure is particularly efficient, providing recovery values ranged from 81 to 95% for both compounds (Table 3). Our validation data were comparable to the published ones and obtained by other methods [20,30].

Figure 1a,b show the chromatograms of a negative blood sample, while Figure 2a,b shows the chromatograms from a blood sample spiked with all the substances studied at a 0.8 µg/mL. As can be seen, all the analytes are well separated and can be identified by their characteristic fragment ions and retention times.

The method was applied to 41 real samples from death subjects submitted to our laboratory for toxicological analysis (Table 4). Samples were selected after a positive result for Δ9-THC in the immunoassay screening procedure in our laboratory.

It is frequently impossible to differentiate occasional from chronic cannabis use when considering a single blood specimen. Thus, while some authors describe a blood concentration of Δ9-THC-COOH of 3 µg/mL as a marker of occasional cannabis intake, others indicate that a concentration of 40 µg/mL can be considered as a marker of near-daily cannabis use [10,36]. In our study, Δ9-THC-COOH was identified in all the subjects at concentrations ranging from 0.12 to 3.38 µg/mL; Δ9-THC could be quantified only in 15 samples. Figure 3a,b shows the chromatograms of sample number 25. The method was demonstrated to be suitable for its application in forensic toxicology.

In the consulted bibliography several articles have been found where Δ9-THC and Δ9-THC-COOH are determined in blood [18,22,23]. Most of them require a first protein precipitation step, which is usually done by adding a certain amount of acetonitrile and then centrifuging for up to 30 min. In the procedure that we describe, this step is not necessary, thereby saving time and ensuring that the extraction can be carried out completely.

In addition, the extraction is carried out by SPE in many of the articles [13,18,22,23] that require a certain time to be able to carry out the different steps of conditioning, washing, and elution. Another factor to consider is the risk of postmortem blood clogging the SPE cartridge, making extraction impossible. 

It is necessary to develop analytical methods that are applicable to forensic samples, mainly postmortem blood, due to the particularities of this type of sample. Real postmortem blood samples are generally difficult to handle due to their poor condition, their low quantity, and often it is not possible to obtain plasma. For this reason, we developed a simpler and faster LLE method adapted to the particularities of this sample and demonstrated its usefulness by analyzing 41 real samples.

## 3. Materials and Methods

### 3.1. Chemicals Reagents and Standards

Hexane, ethyl acetate, sodium hydroxide, hydrochloric acid, acetic acid, methanol, BSTFA, and TMCS were obtained from Merck^®^ (Madrid, Spain). Δ9-tetrahydrocannabinol (Δ9-THC), 11-nor-9-carboxy-Δ9-tetrahydrocannabinol (Δ9-THCCOOH), and their deuterated analogues Δ9-THC-D3 and Δ9-THCCOOH-D3 all of them 100 µg/mL, were obtained from Cerilliant^®^ (Round Rock, TX, USA). Distilled water was processed through a Milli-Q water system (Millipore, Bedford, MA, USA).

### 3.2. Specimens

To carry out the addition curves, drug-free blood was obtained from the Galician Organs and Blood Donation Agency (ADOS). Postmortem whole blood was collected according to the routine autopsy procedures from the Galician Legal Medicine Institute (IMELGA) and sent to our laboratory for toxicological analysis.

Individual methanolic stock solutions containing 1, 10, and 100 µg/mL were used to prepare the spiked blood at concentrations of 0.05, 0.1, 0.2, 0.5, 0.8, 1, and 1.5 µg/mL for Δ9-THC and 0,08, 0.1, 0.2, 0.5, 0.8, 1, and 1.5 µg/mL for Δ9-THC-COOH. Stock and working blood samples as well as methanolic standard solutions were stored at −20 °C until use.

### 3.3. Analytical Procedure

Sample pretreatment and liquid-liquid extractionBlood plasma was separated by centrifugation of whole blood when it was possible, and 1 mL of plasma was placed in a screw-capped round bottom glass tube. First, 20 µL methanolic solution of 10 µg/mL of each internal standard (Δ9-THC-D3 and Δ9-THC-COOH-D3) were added. The tube was closed, and its contents mixed. Afterwards, 1 mL of Milli-Q water and 500 µL of 2 M NaOH is added, and the mixture is again stirred. Subsequently, a liquid-liquid extraction was performed with 5 mL of a mixture (9:1, *v*/*v*) of hexane-ethyl acetate.The tube is agitated for 10 min to cause the migration of components in the two phases, then it is centrifuged for 10 min, and the organic phase containing the THC is transferred and evaporated to dryness. The dried residue was reconstituted with 40 µL of methanol and injected in the GC-MS system.In order to determine Δ9-THC-COOH, a second extraction is needed starting from the aqueous phase.1 mL of a 0.1 N HCl solution and 200 mL of an acetic acid solution were added, in order to acidify the mixture. Then, it was mixed by agitation for 30 s, and 5 mL of the hexane-ethyl acetate mixture (9:1, *v*/*v*) was added again. The tube is agitated for 10 min, centrifuged, and the organic phase transferred to a conical bottom tube with cap and evaporated for later derivatization.The derivatization is carried out by adding a mixture of 40 µL of BSTFA-TCMS (99:1) to the Δ9-THC-COOH dry residue and incubating at 100 °C for 20 min.GC-MSGC-MS analyses were performed in an Agilent 6890 Gas Chromatograph equipped with a 7683B automatic liquid sampler, coupled with an Agilent 5973 mass selective quadrupole detector (Agilent Technologies, Las Rozas, Madrid, Spain). The GC injection port was set at 250 °C in splitless mode (purge time 0.75 min). The GC was equipped with an Agilent 19091S-133U 5% phenylmethylsyloxane capillary column, 30 m × 0.25 mm. i.d., 0.50 µm film thickness (purchased by Agilent Technologies, Las Rozas, Madrid, Spain). The oven temperature was held at 90 °C for 1 min, then at 35 °C/min to 200 °C, then at 10 °C/min to 260 °C and held 15 min. Helium was used as carrier gas at a flow of 1 mL/min. The mass detector operated in electron ionization at 70 eV. Initially, a mixture of standards of all the compounds was analyzed in full scan mode (mass range 50–550 amu). Quantifier and qualifier ions used for each analyte were selected based on their abundance and mass-to-charge ratio (*m*/*z*). Owing to their reproducibility and lack of interferences, high-mass ions were selected whenever possible. Upon the selection of ions, the mass analyzer was operated in selected ion monitoring (SIM) acquisition mode. All diagnostic ions and retention times are listed in Table 5.

### 3.4. Method Validation

Limits of detection, lower limit of quantitation, and specificityThe sensitivity of the method was determined by the calculation of the limit of detection (LOD) and the lower limit of quantitation (LLOQ). LOD was determined by an empirical method that consists of analyzing a series of plasma samples containing decreasing amounts of the analytes. LOD was the lowest concentration that presented a S/N > 3 for at least three diagnostic ions for each substance. The LLOQ was the lowest concentration of analytes in a sample that can be determined with appropriate precision and accuracy. Specificity was studied analyzing 10 negative plasma samples.LinearityThe linearity of the method for each compound was studied in the range 0.05–1.5 µg/mL for Δ9-THC and 0.08–1.5 µg/mL for Δ9-THC-COOH, performing 5 extractions and analyses for each level. Calibration curves were built by linear regression of the area ratio of each substance with the internal standard (IS) vs. the concentration of each analyte. Curves with a quadratic regression coefficient (R^2^) higher than 0.99 were satisfactory.Precision and accuracyPrecision, expressed as the coefficient of variation (CV) of the measured values, was expected to be less than 15% at all concentrations, except for the LLOQ for which 20% was acceptable [27]. It was studied on 5 replicate analyses at three levels: 0.1, 0.5, and 1 µg/mL. In the same way, accuracy was evaluated using the mean relative error (MRE), which had to be less than 15% of the theoretical values at each concentration level except for the LLOQ, for which 20% was acceptable [37].RecoveryThe recovery of an analyte is the detector response obtained from an amount of the analyte added to and extracted from the biological matrix compared to the detector response obtained for the true concentration of the pure authentic standard [38]. Recovery of the analyte must be optimized to ensure that the extraction is efficient and reproducible. Recovery does not need to be 100%, but the degree of recovery of an analyte and internal standard must be consistent and reproducible. The recovery of the method was examined by comparing the analytical results for extracted samples at 2 levels of concentration (0.1 and 1 µg/mL) 5 times within 3 days versus samples spiked with the standards after the extraction step, where unextracted standards represent 100% recovery.

## 4. Conclusions

A fast and sensitive GC-MS method is described for the determination and quantitation of Δ9-THC and Δ9-THC-COOH in postmortem blood samples involving a liquid-liquid extraction procedure.

The proposed method has been validated complying with all the parameters required by the FDA. The method is specific, linear, and precise, being suitable for use in the routine analysis.

The use of a liquid-liquid extraction method provides a low-cost, easy-to-perform, and high-recovery technique. For this reason, it is easy to apply to the reality of the majority of forensic toxicology laboratories with fewer resources since it does not require specialized equipment or a high cost. Although there are currently more sophisticated techniques that achieve greater sensitivity, they are more expensive and are therefore difficult to apply in many laboratories.

On the other hand, the choice of the sample used in this work (postmortem blood) is not the usual one in existing studies, and yet, it is one of the most frequently received samples in forensic toxicology laboratories.

## Figures and Tables

**Figure 1 molecules-28-03586-f001:**
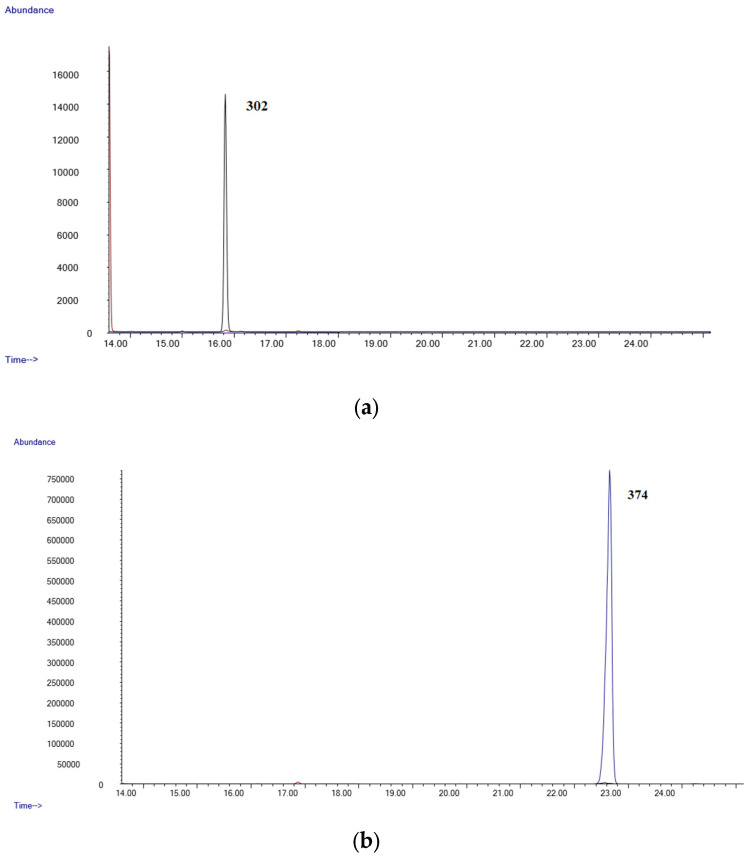
(**a**) Chromatogram of a blank blood sample (Δ9-THC); (**b**) chromatogram of a blank blood sample (Δ9-THC-COOH).

**Figure 2 molecules-28-03586-f002:**
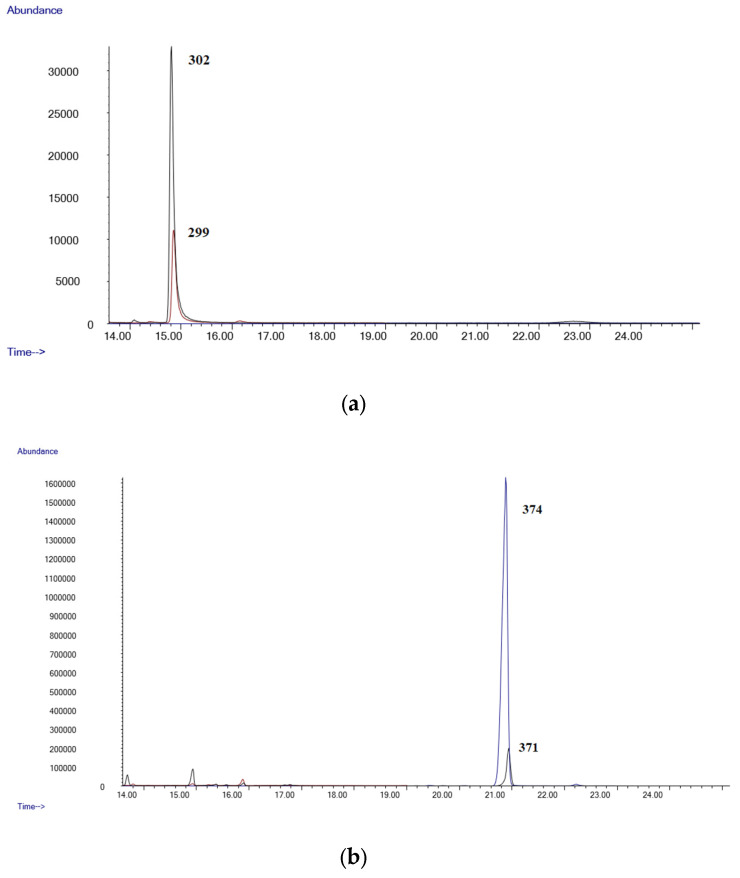
(**a**) Chromatogram of a spiked blood sample (Δ9-THC 0.8 µg/mL); (**b**) chromatogram of a spiked blood sample (Δ9-THC-COOH 0.8 µg/mL).

**Figure 3 molecules-28-03586-f003:**
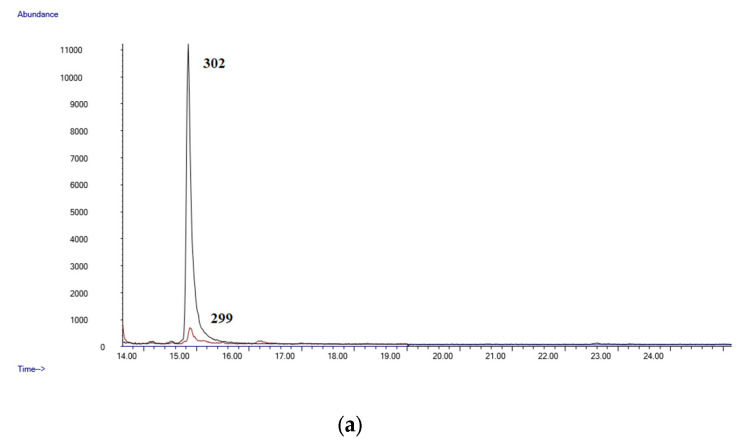
(**a**) Chromatogram of a postmortem blood sample (Case 25, Δ9-THC: 0.11 µg/mL); (**b**) chromatogram of a postmortem blood sample (Case 25 Δ9-THC-COOH: 1.77 µg/mL).

**Table 1 molecules-28-03586-t001:** Limit of detection, lower limit of quantitation, and calibration results for Δ9- THC and Δ9-THC-COOH in plasma.

	LOD (µg/mL)	LLOQ (µg/mL)	Slope	Intercept	R^2^ Coef
Δ9-THC	0.02	0.05	4.455	−0.474	0.991
Δ9-THC-COOH	0.04	0.08	1.754	−0.234	0.992

**Table 2 molecules-28-03586-t002:** Precision and accuracy obtained for Δ9-THC and Δ9-THC-COOH.

		Intraday (*n* = 5)		Interday (*n* = 5)	
	Concentration (µg/mL)	CV (%)	Relative Mean Error (%)	CV (%)	Relative Mean Error (%)
Δ9-THC	0.1	0.43	10.14	5.71	13.45
	0.5	1.08	0.65	10.30	10.10
	1	2.12	7.74	1.13	−3.13
Δ9-THC-COOH	0.1	12.92	−8.78	13.40	10.41
	0.5	1.75	12.48	12.36	−4.51
	1	6.58	13.80	8.87	−0.12

**Table 3 molecules-28-03586-t003:** Recovery data obtained for Δ9-THC and Δ9-THC-COOH.

		Intraday (*n* = 5)		Interday (*n* = 5)	
	Concentration (µg/mL)	Mean Recovery (%)	CV (%)	Mean Recovery (%)	CV (%)
Δ9-THC	0.1	82.95	10.22	83.54	5.79
	1	81.51	7.28	80.69	8.28
Δ9-THC-COOH	0.1	95.41	3.83	94.21	4.18
	1	81.01	9.81	86.12	6.82

**Table 4 molecules-28-03586-t004:** Details of real cases testing positive for Δ9-THC and Δ9-THC-COOH.

Case Number	Gender	Age	Cause of Death	[Δ9-THC] (µg/mL)	[Δ9-THC-COOH] (µg/mL)	Other Detected Substances
1	M	37	Traffic Accident	<LLOQ	0.26	-
2	M	48	Drug Overdose	-	0.28	Methadone, Benzodiacepines
3	M	50	Natural Death	-	0.30	
4	M	42	Traffic Accident	-	0.12	Ethanol
5	M	42	Natural Death	-	0.14	Ethanol, Paracetamol
6	M	23	Suicide. Hanging	-	0.22	-
7	F	45	Natural Death	-	0.83	-
8	M	23	Traffic Accident	0.11	0.37	Ethanol
9	M	40	Drug Overdose	-	0.26	Cocaine, Benzoilecgonine
10	M	53	Natural Death	0.12	0.53	Methadone
11	M	46	Suicide. Hanging	-	0.17	-
12	M	52	Drug Overdose	-	0.19	Ethanol, Methadone, Benzodiacepines
13	M	43	Natural Death	0.11	0.83	-
14	M	57	Drowning	-	0.26	-
15	M	61	Drowning	-	0.32	-
16	M	40	Drug Overdose	-	0.20	Benzoilecgonine, Citalopram
17	M	48	Drug Overdose	-	0.38	Ethanol, Cocaine, Benzoilecgonine, Methadone
18	F	37	Drowning	0.11	0.28	-
19	M	40	Drug Overdose	0.12	0.22	Cocaine, Benzoilecgonine
20	M	35	Suicide. Intoxication	-	0.16	Venlafaxine, Cocaine, Benzoilecgonine, Bupropion
21	M	60	Gas Intoxication	0.14	3.26	Carboxyhemoglobin
22	M	53	Natural Death	-	0.40	-
23	M	19	Traffic Accident	-	3.11	-
24	M	37	Suicide. Precipitation	-	3.38	Ethanol, Cocaine Benzoilecgonine
25	M	47	Drug Overdose	0.11	1.77	Cocaine, Benzoilecgonine
26	M	48	Drug Overdose	-	0.65	Ethanol, Cocaine, Benzoilecgonine
27	M	35	Drug Overdose	0.13	1.14	Ethanol, Methadone, Gabapentine
28	M	23	Traffic Accident	-	0.43	Ethanol
29	F	22	Suicide. Hanging	0.12	0.25	
30	M	39	Natural Death	0.11	0.32	
31	F	18	Traffic Accident	0.13	1.15	-
32	M	59	Drug Overdose	-	0.15	Benzodiacepines, methadone, chlometiazole, trazodone
33	M	38	Suicide. Hanging	-	0.36	Cocaine and metabolites
34	F	43	Drug Overdose	0.12	0.16	Cocaine and metabolites, methadone
35	M	47	Traffic Accident	-	0.37	-
36	F	18	Suicide. Precipitation	0.12	0.25	Ethanol
37	F	38	Traffic Accident. Pedestrian	-	0.20	Ethanol, Cocaine and metabolites
38	M	39	Drug Overdose	-	0.50	Cocaine and metabolites
39	F	42	Stabbed	-	0.32	Ethanol
40	M	60	Natural Death	0.17	0.21	-
41	M	44	Natural Death	-	0.22	-

M: male; F: female. Samples that exceed the range of validation have been diluted in order to quantify them.

**Table 5 molecules-28-03586-t005:** Retention times and ions selected for monitorization.

	Retention Time (min)	Quantifier Ion (*m*/*z*)	Qualifiers Ions (*m*/*z*)
THC	14.80	299	271, 314
THC-D_3_	14.80	302	274, 317
THC-COOH	20.4	371	473, 488
THC-COOH-D_3_	20.4	374	476, 491

## Data Availability

The data that support the findings of this study are available on request from the corresponding author.

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
