# Peer review of "Simple Method for the Determination of THC and THC-COOH in Human Postmortem Blood Samples by Gas Chromatography—Mass Spectrometry"

_molecules, 2023, doi:10.3390/molecules28083586_

Round 1

Reviewer 1 Report

The manuscript entitled ‘SIMPLE METHOD FOR THE DETERMINATION OF THC AND THC-COOH IN HUMAN POSTMORTEM BLOOD SAMPLES BY GAS CHROMATOGRAPHY/MASS SPECTROMETRY’ is well-written and well-organized. The paper is not quite innovative; however, GC/MS is a well-established method in forensic laboratories and is still very interesting. The authors developed a method for THC and THC-COOH in postmortem blood specimens by GC-MS and the results from validation study are acceptable.

The main and very important problem of the manuscript, is the very limited application in forensic cases, because of the high LOD and LOQ of the developed method (0.02 and 0.05 µg/mL for THC respectively). Its applicability of the method is limited only in those cases with THC concentrations in blood above 0.05 µg/mL and that is the problem. For example, for many countries the blood THC limit in traffic accidents is 1, 2 or 5 ng/ml, and it is not reasonable to consider e.g. a THC level of 0.01 µg/mL in blood as negative.   

For this reason the manuscript is not suitable for publication.

Author Response

- The manuscript entitled ‘SIMPLE METHOD FOR THE DETERMINATION OF THC AND THC-COOH IN HUMAN POSTMORTEM BLOOD SAMPLES BY GAS CHROMATOGRAPHY/MASS SPECTROMETRY’ is well-written and well-organized. The paper is not quite innovative; however, GC/MS is a well-established method in forensic laboratories and is still very interesting. The authors developed a method for THC and THC-COOH in postmortem blood specimens by GC-MS and the results from validation study are acceptable.

- The main and very important problem of the manuscript, is the very limited application in forensic cases, because of the high LOD and LOQ of the developed method (0.02 and 0.05 µg/mL for THC respectively). Its applicability of the method is limited only in those cases with THC concentrations in blood above 0.05 µg/mL and that is the problem. For example, for many countries the blood THC limit in traffic accidents is 1, 2 or 5 ng/ml, and it is not reasonable to consider e.g. a THC level of 0.01 µg/mL in blood as negative.

Answer: The authors wish to thank the reviewer for his comment.

The reviewer is correct that the blood limits for drivers are lower than those proposed in this paper, but in this case, the results refer to postmortem blood. In our laboratory, all cases of forensic involvement that arrive are analyzed and not only traffic accidents. On the other hand, in our country there are no non-punishable limits of THC consumption in cases of traffic accidents, and any concentration of THC detected is illegal in the case of drivers.

 Regarding other published works, we are aware of the differences found in the detection and quantification limits, but it is necessary to take into account that many of these works have two-dimensional GC-MS, LC-MSMS and GC-MSMS equipment that is more sophisticated. , sensitive and especially more expensive than the equipment used in this work. As it is a work carried out on postmortem blood, such low limits are not needed since the concentrations that are usually found in deceased people are usually much higher.

Reviewer 2 Report

Manuscript ID: molecules-2290947

Type of manuscript: Article
Title: SIMPLE METHOD FOR THE DETERMINATION OF THC AND THC-COOH IN HUMAN POSTMORTEM BLOOD SAMPLES BY GAS CHROMATOGRAPHY /MASS SPECTROMETRY Authors: Ivan Alvarez-Freire , Anxa Valeiras-Fernández , Pamela Cabarcos Fernández , Ana María Bermejo Barrera , María Jesús Tabernero-Duque*

Special Issue Forensic Analysis in Chemistry

In this article, it proposes a simple and sensitive analytical method for determination of Δ9-tetrahydrocannabinol (Δ9-THC) and its metabolite 11-nor-Δ9-tetrahydrocannabinol-carboxylic acid (Δ9-THC-COOH) in blood sample of human postmortem using GC/MS (SIM) mode.

The authors describe the analytical method as simple and sensitive but any details about that is giving for support that.

In the same way, a method for the simultaneous determination of a psychoactive compound Δ9-THC and a metabolite Δ9-THC-COOH is described, founding some articles about it (around eight in PUBMED) previously described in the literature.

The authors should answer the following questions:

-    What innovation does this methodology really bring to previously published articles?

-    In which part of the procedure specifically is the method simplified?

-    When the authors talk about sensitivity, what terms do they do in? so, taking into account the slope of the calibration line or in terms of LOD and LOQ values? Data and discussion of this, need to be included both in the abstract and in the introduction.

The method involved a liquid-liquid extraction in two steps, one for THC using THC‐D3 as IS and the second extract for THC-COOH was derivatized and analyzed using THC‐COOH‐D3 as IS.

-    How and why does proposing two steps in the extraction process make simple and rapid the process?

Two steps in the LLE process means two extracts and two different runs to analyse just two compounds, is that correct?

Different extraction processes have been previously described in the literature, being LLE the most traditional method.

-    Why the authors use LLE versus other extraction process? What advantages does it have?

Authors should discuss both issues in the introduction.

The last paragraph of the introduction includes more than 15 articles on previously published analytical methods for determination of Δ9-THC and/or its metabolites in body fluids, but they are not date up and do not discuss what the proposed new methodology brings to the previously published papers.

For examples, references 19 y 33 are articles whose proposing the determination of the analytes under study using LLE and GC-MS.

Is the GC-MS method a new methodology or is it reproduced based on a previously published method? If so, it should be cited, and if not, , the authors should in the Results and Discussion section, explain all about the optimisation of the instrumental parameters, both chromatographic and mass spectrometric, has consisted of.

In this sense, Table 1 are results and should be discussed if different studies have been developed.

Having reviewed the entire paper, under my view, this work has a strong point but it is related to the type and number of samples used in the application, more than to quality or innovation of the analytical methodology proposed. However, there is no emphasis on that, neither in the abstract nor in the introduction.

Therefore, it is advised that the authors review the orientation of the paper and rewrite it taking into account the most relevant aspects of the paper.

Minor suggestions:

-    Units followed by data must be separated by a space:  example –20 ºC or 100 %

-    GC/MS à GC-MS

-    Regression coefficient R2

-   

Author Response

- In this article, it proposes a simple and sensitive analytical method for determination of Δ9-tetrahydrocannabinol (Δ9-THC) and its metabolite 11-nor-Δ9-tetrahydrocannabinol-carboxylic acid (Δ9-THC-COOH) in blood sample of human postmortem using GC/MS (SIM) mode.

The authors describe the analytical method as simple and sensitive but any details about that is giving for support that.

In the same way, a method for the simultaneous determination of a psychoactive compound Δ9-THC and a metabolite Δ9-THC-COOH is described, founding some articles about it (around eight in PUBMED) previously described in the literature.

Answer: The method is described as simple and sensitive because although it is true that there are some articles on the determination of THC and THCCOOH, most of these publications use methods that use SPE as the extraction technique, which implies a greater economic expense and more time in the extraction process. Although several articles on the determination of both compounds can be found in PUBMED, many perform the determination in urine or other samples, and not in postmortem  blood.

 -    What innovation does this methodology really bring to previously published articles?

Answer: Despite the fact that in the consulted bibliography, as we have just mentioned, there are several works where THC and its metabolite THCCOOH are determined in blood, several of these works carry out the determination by SPE.

SPE is a very selective technique, requires the use of various solvents (conditioning, washing and elution) and the purchase of the SPE cartridges, which are not usually cheap. We believe that our proposal makes the extraction procedure cheaper and easier, so the method is described as simple.

-    In which part of the procedure specifically is the method simplified?

Answer: When carrying out the extraction of the sample with the proposed procedure, the two analytes can be extracted with the same amount of sample. Previously, in our laboratory, two different aliquots were necessary, one for each analyte. It must be taken into account that in a forensic toxicology laboratory, often the amount of sample with which we can work is scarce and sample savings are always essential, apart from simplifying the extraction process.

  • When the authors talk about sensitivity, what terms do they do in? so, taking into account the slope of the calibration line or in terms of LOD and LOQ values? Data and discussion of this, need to be included both in the abstract and in the introduction.

Answer: As stated in the manuscript on Page 5 line 181, the sensitivity is expressed in terms of LOD and LOQ as indicated by the validation guidelines followed (FDA). We have described the validation parameters in the methodology section of the study and it has not been described in the introduction, since the publication instructions of Molecules indicate that the introduction should refer to the study in a broad context and highlight the purpose, the importance of the work and its meaning.

- The method involved a liquid-liquid extraction in two steps, one for THC using THC‐D3 as IS and the second extract for THC-COOH was derivatized and analyzed using THC‐COOH‐D3 as IS.    How and why does proposing two steps in the extraction process make simple and rapid the process?

Answer: The simplicity of the method is found fundamentally in the use of a single sample to determine the two compounds, which saves time and, above all, the amount of sample. In addition, as we mentioned before, although there are other publications on the determination of THC and THCCOH, they also do it with two elutions steps. Due to the nature of the compounds analyzed, an extraction in a basic medium for THC and in an acid medium for THC-COOH is necessary. Moreover we consider that the liquid-liquid extraction is faster than the SPE.

- Two steps in the LLE process means two extracts and two different runs to analyse just two compounds, is that correct?

Answer: Yes, it is correct (as we just explained in the previous answer)

- Different extraction processes have been previously described in the literature, being LLE the most traditional method. Why the authors use LLE versus other extraction process? What advantages does it have? Authors should discuss both issues in the introduction.

Answer: In a forensic toxicology laboratory, the main advantage of LLE is that it is possible to work with postmortem blood directly. Sometimes the blood degrades and introducing it into the SPE cartridges causes them to get stuck, making extraction difficult or impossible.

This sentence has been added to the introduction as the reviewer suggested.

- The last paragraph of the introduction includes more than 15 articles on previously published - analytical methods for determination of Δ9-THC and/or its metabolites in body fluids, but they are not date up and do not discuss what the proposed new methodology brings to the previously published papers.

For examples, references 19 y 33 are articles whose proposing the determination of the analytes under study using LLE and GC-MS.

Answer:The article in reference 19 uses two-dimensional gas chromatography–mass spectrometry and the article of reference 33 is made in urine and not in postmortem blood, but as suggested by the reviewer, we have added the following paragraph in the discussion:

In the consulted bibliography several articles have been found where THC and THCCOOH are determined in blood. Most of them require a first protein precipitation step, which is usually done by adding a certain amount of acetonitrile and then centrifuging for up to 30 minutes. In the procedure that we describe this step is not necessary thereby saving time and ensuring that the extraction can be carried out completely.

In addition, the extraction is carried out by SPE in many of the articles (13, 18, 22-24) that requires a certain time to be able to carry out the different steps of conditioning, washing and elution. Another factor to take into account is the risk of postmortem blood clogging the SPE cartridge, making extraction impossible.

For all the above, the proposed LLE method is simpler and faster than most of the other methods published

- Is the GC-MS method a new methodology or is it reproduced based on a previously published method? If so, it should be cited, and if not, the authors should in the Results and Discussion section, explain all about the optimisation of the instrumental parameters, both chromatographic and mass spectrometric, has consisted of. In this sense, Table 1 are results and should be discussed if different studies have been developed.

Answer: The described method is the one used routinely in our laboratory, although we have never published it. The method was optimized and validated as described in section 2. Material and Methods, in 2.1.3. Analytical procedure. Table 1 reflects the retention times and the selected ions. The ion with the highest abundance was selected as the quantifier ion. Qualifiers ions were chosen from the next most abundant after the quantifier, as long as they did not coincide with the ions of their corresponding isotope labeled internal standards.

- Having reviewed the entire paper, under my view, this work has a strong point but it is related to the type and number of samples used in the application, more than to quality or innovation of the analytical methodology proposed. However, there is no emphasis on that, neither in the abstract nor in the introduction.

Answer: The importance of applying our method to real samples has been added both in the abstract and in the introduction, emphasizing the particularities of postmortem blood, and in the discussion, explaining the importance of applying our method to a significant number of real samples (41).

- Minor suggestions:

-    Units followed by data must be separated by a space:  example –20 ºC or 100 %

-    GC/MS à GC-MS

-    Regression coefficient R2

Answer: Changes have been made throughout the manuscript

Reviewer 3 Report

This paper describes a development of a quantitative method to determine THC and its metabolite THC-COOH in human postmortem blood. The developed method was applied to human postmortem plasma samples using isotope labeled internal standards, LLE, and GC/MS. Although this study is interesting and would be helpful to researchers in this research field, this paper included several defects. Therefore, to publish on Molecules, the reviewer thinks that the authors should revise this manuscript substantially. The following points would be helpful to improve this paper.

1. In Introduction, from first to third paragraph, the authors described various background knowledge, but cited only one reference. Please cite appropriate reference on its corresponding sentence. Furthermore, according to Reference number 1, the authors accessed the report on 10th January 2020, please re-accessed on recent date and please update access date.

2. In addition to the above opinion, the reviewer thinks that, in Introduction section, although the second paragraph and the third paragraph contained similar information, the authors provided different values (such as 55.4 million males and 36.1 million females for cannabis use in second paragraph, but 91.2 million adults for cannabis use in third paragraph). Moreover, the reviewer thinks that these 2 paragraphs are redundant and the authors could condense and re-arrange overall Introduction section.

3. In overall manuscript, if the authors use an abbreviation for full name, the authors provide both full name and its corresponding abbreviation at first, after that should only use the abbreviation. And the authors should provide full name for abbreviation before use abbreviations. Furthermore, the authors used different abbreviations for same full name. Please use one abbreviation for same name.

4. In overall manuscript, please check superscripts and subscripts.

5. In Materials and Methods, on 2.1.3. Analytical procedure section, the authors used dots instead of using sub-section number. If the authors followed according to authors’ guidelines of ‘Molecules’, the authors can ignore this comment.

6. In Results and Discussion, the authors described CBD for useful markers for its potential therapeutic used. However, the reviewer thinks that the description for CBD did not harmonize context of this manuscript. Please delete those sentences and rewrite.

7. On Table 2, the authors use R coef. However, in 2.1.3. Analytical procedure, the authors described that the squared R for linearity of the method. Which one is correct? Please revise.

8. Figures 1 and 2 occupied too large space. The reviewer thinks that those figures could be condensed one figure with re-sizing.

9. On line 303, the authors described Figure 3 shows the chromatograms for sample number 1. However, on figure caption, the authors wrote ‘Case 25’. Which one is correct? Please revise.

Author Response

This paper describes a development of a quantitative method to determine THC and its metabolite THC-COOH in human postmortem blood. The developed method was applied to human postmortem plasma samples using isotope labeled internal standards, LLE, and GC/MS. Although this study is interesting and would be helpful to researchers in this research field, this paper included several defects. Therefore, to publish on Molecules, the reviewer thinks that the authors should revise this manuscript substantially. The following points would be helpful to improve this paper.

  1. In Introduction, from first to third paragraph, the authors described various background knowledge, but cited only one reference. Please cite appropriate reference on its corresponding sentence. Furthermore, according to Reference number 1, the authors accessed the report on 10th January 2020, please re-accessed on recent date and please update access date.

 Answer: It was an error on our part to use a previous report. We have accessed the last published report and changed the date in the manuscript. Quotes number 1 and 2 correspond to the data supplied by the EMCDDA Report, from different years. The paragraphs with the information on these data have been reformulated, and the citations have been updated.

  1. In addition to the above opinion, the reviewer thinks that, in Introduction section, although the second paragraph and the third paragraph contained similar information, the authors provided different values (such as 55.4 million males and 36.1 million females for cannabis use in second paragraph, but 91.2 million adults for cannabis use in third paragraph). Moreover, the reviewer thinks that these 2 paragraphs are redundant and the authors could condense and re-arrange overall Introduction section.

  Answer: We have modified both paragraphs, updating the data according to the most recent Report. We have also condensed the information, reducing introductory paragraphs.

  1. In overall manuscript, if the authors use an abbreviation for full name, the authors provide both full name and its corresponding abbreviation at first, after that should only use the abbreviation. And the authors should provide full name for abbreviation before use abbreviations. Furthermore, the authors used different abbreviations for same full name. Please use one abbreviation for same name.

  Answer: These abbreviations has been changed

  1. In overall manuscript, please check superscripts and subscripts.

   Answer: The superscripts and subscripts has been checked

  1. In Materials and Methods, on 2.1.3. Analytical procedure section, the authors used dots instead of using sub-section number. If the authors followed according to authors’ guidelines of ‘Molecules’, the authors can ignore this comment.

    Answer: For editing the document, we follow the guidelines provided by the journal Molecules.

  1. In Results and Discussion, the authors described CBD for useful markers for its potential therapeutic used. However, the reviewer thinks that the description for CBD did not harmonize context of this manuscript. Please delete those sentences and rewrite.

     Answer: The sentences were eliminated from the manuscript

  1. On Table 2, the authors use R coef. However, in 2.1.3. Analytical procedure, the authors described that the squared R for linearity of the method. Which one is correct? Please revise.

 Answer: We're sorry. It was a mistake that has now been fixed. The correct one is R squared.

  1. Figures 1 and 2 occupied too large space. The reviewer thinks that those figures could be condensed one figure with re-sizing.

 Answer: At first we had tried to do what the reviewer indicated but the information in the final image was not clear and the information could not be read properly

  1. On line 303, the authors described Figure 3 shows the chromatograms for sample number 1. However, on figure caption, the authors wrote ‘Case 25’. Which one is correct? Please revise.

Answer: The correct is the sample number 25. It has been corrected in the text. Thank you for finding this mistake

Reviewer 4 Report

Title must be changed: their method is not simple while they used derivatized agent and two different extraction method.

Introduction lack of refences introduction must rewritten.

objective of this work is not clear and should be place at the end of introduction.

Why they did not analyze 11-hydroxy-THC and CBD.

Method section.

Long run time, analytes appeared at 18 min and after 21 min. 

line 190-192: The linearity of the method for each compound was studied in the range 0.05–1.5 190 µg/ml for THC and 0.08-1.5 µg/ml for THC-COOH'' why they chose this range. In most previous report, a higher level was chosen.

line 194: change... regression coefficient to coefficient of determination.

Have they tested dilution? 

Results and discussions:

Most of method description, sensitively, selectivity... etc. can be moved to method section. 

It is recommended that results to be separated from discussions. 

Author Response

- Title must be changed: their method is not simple while they used derivatized agent and two different extraction method.

Answer: The method is described as simple and sensitive because the two analytes can be extracted with the same amount of sample. Previously, in our laboratory, two different aliquots were necessary, one for each analyte. It must be considered that in a forensic toxicology laboratory, the amount of sample with which we can work is often scarce and sample savings are always essential, in addition to simplifying the extraction process.

The use of a single sample to determine the two compounds saves time and, above all, amount of sample. Furthermore, most publications on THC and THCCOH determination use SPE and do so with two elution steps. Due to the nature of the compounds analyzed, an extraction in a basic medium is necessary for THC and in an acid medium for THC-COOH. It must also be taken into account that liquid-liquid extraction is faster than SPE.

Although derivatization is necessary to determine this type of molecules with GC-MS, in the case described this is a quick process (20 minutes), while most published articles use a longer derivatization process.

- Introduction lack of refences introduction must rewritten.

Answer: The introduction has been rewritten and references have been added as suggested by the reviewer.

- Objective of this work is not clear and should be place at the end of introduction.

Answer: As suggested by the reviewer, a paragraph has been added at the end of the introduction describing the aim of the paper.

- Why they did not analyze 11-hydroxy-THC and CBD.

Answer: Cannabis used as a drug of abuse is high in Δ9-THC, while legal cannabis contains mostly CBD (and less than 1% Δ9-THC). The determination of 11-nor-9-carboxy-THC (Δ9-THC-COOH) in forensic toxicology laboratories is of special interest, to distinguish between the use of legal cannabis (mainly with CBD) and that of cannabis as a drug of abuse (mainly Δ9-THC, from which the inactive metabolite Δ9-THC-COOH is derived). This last situation (use as a drug of abuse) is the most frequent in cases with forensic involvement.

THC is the main psychoactive component of cannabis. After smoking, it is rapidly metabolized to 11-OH-THC, which is also psychoactive and short-lived. Subsequently, 11-OH-THC is metabolized to THC-COOH, which is the main inactive metabolite. THC blood concentrations peak ten minutes after smoking cannabis and are rapidly distributed. However, THC-COOH blood concentrations are detectable up to a month after sustained abstinence in chronic users. For this reason, THC-COOH is used for forensic purposes.

- Method section.

- Long run time, analytes appeared at 18 min and after 21 min.

 Answer: Although it is possible to adjust the temperature of the chromatographic oven so that the analytes elute a little earlier, this implies shortening the half-life of the chromatographic column, which is why it has been decided to maintain this temperature ramp with the times described.

- line 190-192: The linearity of the method for each compound was studied in the range 0.05–1.5 µg/ml for THC and 0.08-1.5 µg/ml for THC-COOH.  Why they chose this range. In most previous report, a higher level was chosen:

 Answer: The range of concentrations has been chosen based on the average concentrations usually detected in the cases received in our laboratory.

- line 194: change... regression coefficient to coefficient of determination:

 Answer: For the validation of the method, the FDA validation guidelines have been used, which talk about regression models. That is why we refer to the regression coefficient

- Have they tested dilution? 

Answer: Yes, in several of the real cases it has been necessary to dilute the sample because they exceeded the ranges used (this is indicated in Table 5).

- Results and discussions:

- Most of method description, sensitively, selectivity... etc. can be moved to method section. It is recommended that results to be separated from discussions. 

Answer: The description of the validation parameters is included in the material and methods section, as indicated in the instructions of the journal (Molecules template).

We have combined the Results and Discussion sections because we believe that this way the results obtained and the comparison with other publications are better understood and explained.

Round 2

Reviewer 4 Report

In this report, the presence of THC and its metabolites THC-COOH in blood obtained from postmortem cases using method of analysis consists of liquid-liquid extraction and GC-MS methods.

In fact, detection THC and its metabolites in postmortem cases is not a new for forensic toxicologists, several methods published in the filed using more comprehensive technology and also in some report using direct detection of THC, THC-COOH and their glucuronide. Although the obtaining of a Lower limit of quantification is crucial as in most cases these analytes presence with very low concentration usually LOQ set as 0.001 µg/mL, using hydrolysis step a measurable THC-COOH can be achieved. The unique of the current study is that they are dealing with deaths due to car accidents, drug overdose, natural deaths etc.. As in previous reports that most dealing with aviation accidents especially those from USA. Authors should present this point clearly or otherwise their work is not presenting any new information to be published. 

I certainly not found any improvement in citation through this article introduction, author's still speaking about lots of information with only single citation.

As this work proposed for postmortem cases, authors should follow method validation that used by most of forensic toxicologist around the word which is "ANSI/ASB Standard 036; American Academy of Forensic Sciences Standards Board 2019. Method Validation in Forensic Toxicology: Colorado Springs, CO, USA, 2019"

Authors mentioned that LLE is much better than SPE which is not true taken into account that SPE is much clean, high recovery and most important selectivity and sensitivity. They can argue that SPE is expensive.

Although their method is robust, but it is considered long, they can mention that as one of method limitation which is accepted.